

# Impact of Western Pacific Subtropical High on Ozone Pollution over Eastern China

**Zhongjing Jiang[1], Jing Li[1], Xiao Lu[2], Cheng Gong[3], Lin Zhang[1], Hong Liao[4]**

Department of Atmospheric and Oceanic Sciences, School of Physics, Peking University, Beijing, China
School of Engineering and Applied Sciences, Harvard University, Cambridge, USA
Institute of Atmospheric Physics, Chinese Academy of Sciences, Beijing, China
Jiangsu Key Laboratory of Atmospheric Environment Monitoring and Pollution Control, Jiangsu Collaborative Innovation Center of Atmospheric Environment and Equipment Technology, School of Environmental Science and Engineering, Nanjing University of Information Science and Technology, Nanjing, China

†Corresponding author: jing-li@pku.edu.cn

**Abstract**

Surface ozone is a major pollutant in Eastern China, especially during the summer season. The formation of surface ozone pollution highly depends on meteorological conditions as generally controlled regional circulation patterns. Here we show that summertime ozone pollution over Eastern China is distinctly modulated by the variability of West Pacific Subtropical High (WPSH), a major synoptic system that



controls the summertime weather conditions of East Asia. Composite and regression
analyses indicate that positive WPSH anomaly is associated with higher than normal
surface ozone concentration over Northern China but lower ozone over Southern China.
We show that this is mainly driven by changes in meteorological variables with stronger
than normal WPSH leading to higher temperatures, stronger solar radiation at the land
surface, lower relative humidity, and less precipitation in Northern China, favoring the
production and accumulation of surface ozone. In contrast, all variables show reverse
changes in Southern China under stronger WPSH. GEOS-Chem simulations reasonably
reproduce the observed ozone changes associated with the WPSH and support the
statistical analyses. Detailed contributions of different processes are quantified through
budget diagnosis, which emphasizes the decisive role of chemistry. Natural emission of
precursors from biogenic and soil sources accounts for ~30% of the total surface ozone
changes.
**Key words:**
Surface ozone, WPSH, meteorological fields, GEOS-Chem, precursor

**1. Introduction**

Surface ozone is a major trace gas in the lower atmosphere. It is produced by
photochemical oxidation of carbon monoxide (CO) and volatile organic compounds



(VOCs), in the presence of nitrogen oxides (NOx=NO+NO$_2$) and sunlight. Not only
does it act as a greenhouse gas but it also exerts detrimental effects on both human
health and the ecosystem (Heck et al., 1983; Tai et al., 2014; Monks et al., 2015; Liu et
al., 2018; Maji et al., 2019). In China, the problem of tropospheric ozone pollution is
severe in most urban areas, such as the North China Plain (NCP), the Yangtze River
Deltas (YRD), and Pearl River Deltas (PRD) (Li et al., 2019; Lu et al., 2018; Silver et
al., 2018; Yin et al., 2019). Typically, surface ozone concentration reaches its peak in
the summer season due to active photochemistry (Wang et al., 2017; Lu et al., 2018).
The summertime daily maximum 8h average (MDA8) ozone concentrations frequently
reach or exceed the Grade Ⅱ national air quality standard of 82 ppbv in NCP (Lu et al.,
2018; Ministry of Environmental Protection of the People's Republic of China (MEP),
2012). Moreover, recent studies showed that surface ozone concentration has exhibited
an increasing trend since 2013 over most parts of China (Li et al., 2019; Lu et al., 2020).

Surface ozone concentration is distinctly influenced by meteorological conditions,
which impact the production, transport, and removal of ozone. For example, solar
radiation changes surface ozone via the effects on photolysis rates as well as on biogenic
emissions. High temperature tends to enhance ozone pollution through stagnant air
masses, thermal decomposition of peroxyacetylnitrate (PAN), and the increase of
biogenic emissions (Fehsenfeld et al., 1992; Guenther et al., 2012; Rasmussen et al.,
2012). Wind speed is generally anticorrelated with surface ozone, indicating the
important role of horizontal wind in pollutant dispersion (Zhang et al., 2015; Gong and



Liao, 2019). Moreover, the variabilities of these meteorological variables are not
independent but interconnected. The synchronous variation of some meteorological
variables can be ascribed to the same synoptic weather pattern, thus increasing efforts
have been devoted to identifying the synoptic weather patterns that enhance ozone
pollution (Gong and Liao, 2019; Liu et al., 2019; Han et al., 2020). For example, Liu et
al. (2019) objectively identified 26 weather types including some led to highly polluted
days and proved that synoptic changes account for 39.2% of the interannual increase in
the domain-averaged $O_3$ from 2013 to 2017. Han et al. (2020) also identified six
predominant synoptic weather patterns over eastern China in summer to examine the
synoptic influence of weather conditions on ozone.

A dominant system that affects the summertime weather pattern in China is the WPSH.
As an essential component of the East Asia summer monsoon, its intensity, shape, and
location control the large-scale quasi-stationary frontal zones in East Asia (Huang et al.,
2018). WPSH can significantly influence the monsoon circulation, typhoon tracks, and
moisture transport (Choi et al., 2019; Gao et al., 2014) and further impact surface ozone
in China. Shu et al. (2016) showed stronger WPSH will increase ozone pollution over
YRD by enhancing the ozone production as well as trapping the ozone in the boundary
layer. Using observations from 2014 to 2016, Zhao et al. (2017) indicated that stronger
WPSH in summer leads to a decrease in surface ozone in Southern China but an
increase in Northern China through statistical analysis. While these studies arrived at
qualitative conclusions, they either focused on a limited region or a short time span, and



both lacked a comprehensive investigation of the mechanisms through model
simulation. Considering the increasingly severe ozone pollution in China, it is desirable
to further investigate this topic systematically.

For this purpose, this study aims to address how and why summertime surface ozone
concentration in Eastern China responds to changes in the WPSH. A joint statistical
analysis and model simulation using the GEOS-Chem is performed to reveal their
relationship as well as to examine changes in the relevant chemical and physical
processes, in order to provide insights into the formation of summertime ozone
pollution in China and to shed light on ozone simulation and prediction.

**2.   Data and methods**
**2.1.   Surface ozone and meteorological data**

Routine daily monitoring of air quality in China became available since 2013, with the
establishment of a national network by the China National Environmental Monitoring
Centre. We obtained hourly surface ozone concentration data of all sites available from
2014 to 2018 from https://quotsoft.net/air/. An *ad hoc* quality control protocol was
developed to remove outliers and invalid measurements (see supplementary
information and Figure S1 for example of outliers). MDA8 was calculated based on the
hourly ozone data. We removed the linear trend of the data and converted the data unit
from $\mu g\ m^{-3}$ into ppbv for further analysis. The following calculation is done for cities



with a longitude greater than 100°E which serves as a boundary for a rough definition
of Eastern China.

Meteorological fields for 2014-2018 was obtained from the Goddard Earth Observing
System Forward Processing (GEOS-FP) database, which is the current operational met
data product from the Global Modeling and Assimilation Office (GMAO). The data is
available     at     http://ftp.as.harvard.edu/gcgrid/data/GEOS_2x2.5/GEOS_FP.     The
meteorological variables used include sea level pressure (SLP), cloud cover (CLDTOT),
solar radiation (SWGDN), 2m temperature (T2M), 10m U wind (U10M), 10m V wind
(V10M), total precipitation (PRECTOT) and relative humidity (RH). These variables
are 1-hour averages except for RH that is 3-hour averages. The hourly data is averaged
into daily means for further analysis.

**2.2.     WPSH index and composite analysis**

Figure 1a shows the multi-year averaged summertime SLP field from 1979 to 2018 and
Figure 1b shows its standard deviation. Although the center of the high-pressure system
is located over the Northeastern Pacific Ocean, it also shows substantial variability over
the West Pacific extending to the east coast of China. This west branch has a significant
impact on the summer weather patterns over Eastern China. Wang et al. (2013) defined
a WPSH index to characterize the change of WPSH intensity. It is calculated as the
mean of 850hPa geopotential height anomaly within the 15-25°N and 115-150°E region





(red box in Figure 1b), where the maximum interannual variability of WPSH in the
Western Pacific Ocean is located. Here we adopted this index to represent the strength
and variability of the WPSH.

Using this WPSH index, we defined three types of WPSH conditions, namely strong,
normal, and weak. Specifically, days with WPSH-index exceeding the 90th percentile
of its distribution are classified as strong WPSH days, the 45th -55th percentile as normal
WPSH days, and those below the 10th percentile as weak WPSH days (Figure 1c).
Therefore, each type has 46 days during the summer from 2014 to 2018. Composite
analysis of observed and simulated surface ozone, meteorological variable as well as
related model processes are performed based on these three types.


**2.3.    GEOS-Chem simulations**

We use the GEOS-Chem chemical transport model (CTM) (Bey et al., 2001; v12.3.2;
http://geos-chem.org ) to verify the responses of surface ozone in Eastern China to
changes of the WPSH and to examine changes in the processes involved. GEOS-Chem
includes a detailed $O_x$-$NO_x$-HC-aerosol-Br mechanism to describe gas and aerosol
chemistry (Parella et al., 2012; Mao et al., 2013). The chemical mechanism follows the
recommendations by the Jet Propulsion Laboratory (JPL) and the International Union
of Pure and Applied Chemistry (IUPAC) (Sander, et al., 2011; IUPAC, 2013).





Photolysis rates for tropospheric chemistry are calculated by the Fast-JX scheme (Bian
and Prather (2002); Mao et al. (2010)). Transport is computed by the TPCORE
advection algorithm of Lin and Rood (1996) with the archived GEOS meteorological
data. Cloud convection is computed from the convective mass fluxes in the
meteorological archive as described by Wu et al. (2007). As for boundary layer mixing,
we used the non-local scheme implemented by Lin and McElroy (2010).

Emissions are configured using the Harvard-NASA Emission Component (HEMCO)
(Keller et al., 2014). Biogenic VOC emissions, including isoprene, monoterpenes, and
sesquiterpenes, are calculated online using the Model of Emissions of Gases and
Aerosols from Nature (MEGAN v2.1, Guenther et al., 2012). Soil NOx emissions are
calculated based on available nitrogen (N) in soils and edaphic conditions such as soil
temperature and moisture (Hudman et al., 2012).

The model is driven by GEOS-FP meteorology fields and runs with 47 vertical levels
and $2° \times 2.5°$ horizontal resolution. The model simulations started from January $1^{st}$ and
ended on August $31^{st}$ for each year during 2014 to 2018, in which the first 5 months
were used as spin-up and June-July-August (JJA) are used for composite analysis.
Anthropogenic emissions were fixed in 2010 after which the MIX emission inventory
stopped updating, so that the differences among the three types of WPSH are solely
caused by the change of meteorology. Because meteorology not only affects the
production and transport of ozone but also significantly impacts the emission of BVOCs



and $NO_x$ from the soil, two important precursors of ozone formation, we also performed
another set of simulations with MEGAN and soil NOx emissions turned off to explore
the contribution of natural emissions. We used ozone levels at the lowest model level
with an average height of 58 m to represent model simulated surface ozone
concentration.

**2.4.    Ozone Budget diagnosis**
The simulated ozone concentration is determined by four processes, namely chemistry,
transport (the sum of horizontal and vertical advection), mixing, and convection. Dry
deposition is included in mixing since we used the non-local PBL mixing scheme.
Budget diagnosis is further performed to quantify their individual contributions. The
GEOS-Chem v12.1.0 or later versions provide budget diagnostics defined as the mass
tendencies per grid cell ($kg\ s^{-1}$) for each species in the column (full, troposphere, or
PBL) related to each GEOS-Chem component (e.g, chemistry). These diagnostics are
calculated by taking the difference in the vertically integrated column ozone mass
before and after chemistry, transport, mixing, and convection component in GEOS-
Chem. Here we use the budget diagnostics in the PBL column and calculated composite
means for each type of WPSH.

**3.  Results**
**3.1.    Observed surface ozone changes associated with WPSH intensity**



We first examine the relationship between observed MDA8 and WPSH-index of all
cities in China. Figure 2a&b (symbols) respectively shows the difference in the
composite mean of observed MDA8 between strong/weak WPSH days and normal
WPSH days. A distinct dipole-like pattern can be observed in Figure 2a, indicating that
during strong WPSH events, surface ozone concentration tends to be higher in Northern
China but lower in Southern China, especially the southeast region. The transition from
positive to negative changes happens around 32°N (Figure 2a), which is then used as
the division between Northern and Southern China in this study. In contrast, Figure 2b,
which shows the composite mean difference between weak and normal WPSH days,
also exhibits a dipole pattern but opposite in sign to that shown in Figure 2a.
Quantitatively, for cities with significant differences (p-value<0.05) in Student's $t$-test,
during strong WPSH days, the average MDA8 increased by 10.7 ppbv (+19%, Figure
2a&c) in Northern China and decreased by 11.2 ppbv (-24%, Figure 2a&c) in Southern
China, Under weak WPSH conditions, the average MDA8 decreased by 10.2 ppbv (-
17%, Figure 2b&d) in Northern China and increased by 4.6 ppbv (+10%, Figure 2b&d)
in Southern China. This dipole change of ozone is also confirmed by a regression
analysis of surface ozone against the WPSH index (Figure 2e), with significant positive
signals over Northern China and negative signals in Southern China.

Composite and regression analysis jointly prove the robustness of the dipole-like ozone
anomaly pattern associated with WPSH variability. It is likely that these changes are
driven by changes in meteorological conditions. Therefore, in Figure 3, we further



examine the differences of major meteorological variables associated with WPSH
intensity.

The change of SLP associated with strong WPSH days clearly shows a positive center
in the Northwest Pacific Ocean and to the east of China coast (Figure 3a). This high-
pressure center induces anti-cyclonic circulation anomalies, which manifest themselves
as southwest wind (10 m) anomalies over Eastern China (Figure 3a). In Northern China,
because the surface winds are blown from the land area in the south (Figure 3a), it
contains less moisture but with higher temperatures. As a result, Northern China
exhibits an increase in temperature (Figure 3k), but decreases in moisture-related
variables including precipitation (Figure 3c), relative humidity (Figure 3e), and cloud
cover (Figure 3g). The decrease in cloud cover increases the near-surface solar radiation
(Figure 3i) and can further change the photochemical reaction rates, which partly
explains the increase of ozone concentration (Jeong and Park, 2013; Gong and Liao,
2019). The air stagnation associated with higher temperatures, less precipitation may
also limit the diffusion and removal of ozone (Lu et al., 2019b; Pu et al., 2017).
Moreover, previous studies showed that ozone is negatively correlated with
precipitation and RH (Jeong and Park, 2013; Zhang et al., 2015). The overall changes
of the meteorological fields in Northern China thus act to enhance surface ozone.

In Southern China, the south winds bring moisture from the ocean surface, providing
ample water vapor for the rain band that forms on the northern boundary of the WPSH



(Sampe et al., 2010; Rodriguez et al., 2019). This results in increased precipitation
(Figure 3c), relative humidity (Figure 3e), and cloud cover (Figure 3g), and reduced
surface shortwave radiation (Figure 3i). The increased precipitation and decreased solar
radiation also help to lower the surface temperature (Figure 3k). The corresponding
ozone concentration change is thus negative and opposite to that in Northern China. In
addition, the transport of ozone-depleted air from the ocean can also dilute surface
ozone.

Under the weak WPSH condition, the high-pressure center in Northwest Pacific is
weaker and shifted slightly southward (Figure 3b). The changes of meteorological
variables mostly show reversed patterns to those under strong WPSH cases, but some
asymmetric features are noticed. For example, solar radiation decreased and total
precipitation increased in Guangdong province, which is contrary to the general solar
radiation enhancement and precipitation reduction in Southern China. However, these
abnormal changes in meteorology well match the observed decrease of ozone in
Guangdong province.

According to the weather anomalies related to WPSH intensity, we summarize two
pathways for ozone changes: (1) the relative changes of solar radiation and the
associated meteorological variables impacting on the chemical formation of ozone; (2)
the transport indicated by wind anomalies serving to enrich or dilute ozone
concentration depending on wind direction.




**3.2.    Simulated WPSH impacts on ozone air quality**


Statistical analysis in Section 3.1 only reveals correlation but not causality. To

investigate whether or not the WPSH-related meteorology changes indeed induce the

dipole-like ozone change pattern, we perform GEOS-Chem simulations from 2014 to

2018 with anthropogenic emissions fixed in 2010. In this way, the model responses are

purely attributed to changes in meteorology.

273

The model's capability in reproducing the spatial-temporal variability of MDA8

concentrations in China is first evaluated by comparing the simulation results from 2014

to 2018 over all Chinese cities with observation (Figure S2). GEOS-Chem captures the

observed seasonal spatial distributions of MDA8 reasonably well. The spatial

correlation coefficients (R) between the observed and simulated seasonal mean MDA8

concentrations for summers from 2014 to 2018 are 0.57, 0.59, 0.70, 0.81, and 0.81

respectively, proving the reliability of GEOS-Chem to represent the variation of ozone

MDA8 concentrations.

282

Figure 2 (filled contours) shows the simulated MDA8 changes during strong/weak

WPSH days with respect to normal days (a&b) and their relative changes (c&d).

Compared with observed changes (symbols), GEOS-Chem model well reproduces the

dipole-like pattern of ozone change, albeit with a slight underestimation especially in



Northern China. By calculating the average changes of simulated ozone concentration
sampled at each city, we find the ozone responses to strong and weak WPSH are quite
symmetric, with the average MDA8 increased by 3.6 ppbv (+6%) in Northern China
and decreased by 7.1 ppbv (-12%) in Southern China during strong WPSH (Figure 2a),
and the average MDA8 decreased by 3.6 ppbv (-6%) in Northern China and increased
by 6.6 ppbv (+11%) in Southern China during weak WPSH (Figure 2b). The slight
underestimation of model results may come from the model's lack of ability in
capturing the peak values of ozone MDA8 (Zhang and Wang, 2016; Ni et al, 2018).

**3.3 Budget diagnosis**

In order to examine and to quantify the chemical and physical processes that lead to the
ozone change, Figure 4 provides the budget diagnostics of chemistry, transport, mixing,
and convection in the PBL column. Chemistry represents the changes in net chemical
production, which is determined by the change of reaction rate and the amount of ozone
precursors. As the photolysis rate and natural precursor emissions are both influenced
by meteorological conditions, the change of chemical production is consistent with the
variation of solar radiation and temperature in Figure 3. Under the strong WPSH
condition, ozone concentrations from chemical production exhibit a tripolar structure,
with increases in Northern China and the southern edge and decreases in the Yangtze
River basin (Figure 4a).





309 Transport represents the change of horizontal and vertical advection of ozone. As wind

310 anomalies associated with strong WPSH (Figure 3a) tend to dilute surface ozone in the

311 south and accumulate ozone in the north, the resulting ozone change exhibits an

312 asymmetric pattern with decreases in most of Southern China and increases over

313 Northern and Northeastern China. The mixing process describes the turbulence

314 diffusion in the boundary layer. Mixing in the whole PBL column represents the total

315 exchange of PBL with the free atmosphere, which shows a roughly reversed pattern to

316 chemistry (Figure 4e). Cloud convection shows a general dipole pattern with positive

317 signals in the north and negative signals in the south. However, the small changes in the

318 absolute value suggest a weak impact via deep convection (Figure 4g). Under weak

319 WPSH conditions, ozone from chemical production significantly increases in the east

320 of Southern China but decreases strongly in Northern and Southwestern China (Figure

321 4b). According to the wind anomalies in Figure 3b, transport tends to minimize the

322 difference induced by chemistry and thus leads to an opposite ozone change (Figure

323 4d). Mixing shows a distinct north-south contrast pattern (Figure 4f). Convection

324 changes slightly in opposite direction in the north and south (Figure 4h). Due to PBL

325 mixing, the total change of these processes (Figure 4i&j) in the PBL column shows a

326 consistent pattern with both the observed and simulated change of surface ozone (Figure

327 2). In general, chemistry (Figure 4a&b) and transport (Figure 4c&d) account for the

328 largest proportions of ozone change than the other two mechanisms (i.e., mixing, Figure

329 4e&f, and convection, Figure 4g&h).




In order to provide a more quantitative evaluation of the contribution of these processes,
in Figure 4k-n, we examine the regionally averaged ozone changes for a North (36.0-
42.0°N, 105.0-117.5°E) and South (26.0-32.0°N, 107.5-120.0°E) region, respectively
defined by the purple and black boxes on Figure 4i&j. It can be seen that the regionally
averaged total ozone change is around $\pm$1-2 kg s$^{-1}$. In all cases except Northern China
under strong WPSH, chemistry appears to be the dominating process, which results in
the largest ozone change and with the same sign as the total change and sometimes can
even exceed the amount of total change. For the Northern China case, transport slightly
outweighs chemistry as the primary factor (Figure 4k). Transport contributes to total
changes either positively or negatively, depending on the ozone concentration gradient
and wind anomalies. It tends to increase ozone when the wind anomalies come from
inland regardless of the direction (Figure 4k&m&n). In contrast, when the wind comes
from the ocean, it serves to reduce surface ozone (Figure 4l). As the mixing process
transports ozone along the vertical concentration gradient, it generally contributes
negatively to the total ozone change and thus counteracts excessive chemical changes
(Figure 4l-n). Convection only induces minor modulation to the total changes, generally
less than $\pm$1 kg s$^{-1}$ and negligible for some cases (Figure 4l&m).

**3.4 The contribution of natural emission of ozone precursor gases**

In the GEOS-Chem simulation, all anthropogenic emissions are fixed so there is no
anthropogenic contribution to the simulated ozone change. However, the emission of


ozone precursor gases from natural sources, primarily biogenic volatile organic
compounds (BVOCs) and soil-released NOx (SNOx), closely respond to meteorology
and further impact the chemical production of ozone which has been identified as the
main driving force of ozone change. Therefore, in this section, we continue to quantify
the contribution of BVOCs and soil NOx emission to the ozone changes with WPSH.

Isoprene (used as a proxy of BVOCs) emissions are strongly correlated with
temperatures and increase rapidly between 15 and 35 ℃ (Fehsenfeld et al., 1992;
Guenther et al., 1993), thus the pattern of their changes with WPSH are highly
consistent with the T2 changes (Figure 5a&b). Intensified WPSH results in 10-40%
increases of BVOCs emissions in Northern China and 10-30% decreases in Southern
China, whereas under weak WPSH conditions, they increase strongly in most parts of
China but with a slight decrease over the Northern China Plain and Northeastern China.
Changes of NOx emission from the soil also exhibit a similar pattern to those of T2.
Their responses to weak WPSH appear to be stronger than BVOCs, with decreases up
to 40% over most of Northern China (Figure 5c&d). As most parts of China are the
high-NOx and VOC-limited regions, the overall decreases of BVOCs and NOx reduce
the ozone concentration.

We further quantify the contribution of BVOCs and soil NOx emissions to the changes
in surface ozone concentration, by comparing simulation results with MEGAN and soil
emissions turned on and off. Figure 6a&b and 6c&d shows the simulated MDA8 ozone



with biogenic and soil NOx emissions on and off respectively. They show similar spatial
patterns but the emission-off case exhibits weaker responses. Figure 6e&f shows their
differences, which represent the MDA8 changes due to the combined effect of BVOCs
and soil NOx emission changes associated with WPSH variation. The precursor-
induced ozone changes are in phase with the total ozone changes in most parts of China
and show a dipole-like pattern. In total, these two factors result in ~±1.3 ppbv MDA8
ozone changes (averaged over all cities), which accounts for around 30% of the total
simulated change. Figure6 g&h and i&j show the contribution of soil NOx and BVOCs
emissions respectively, from which we can see that the ozone change induced by soil
$NO_x$ is weaker, implying that BVOCs is the dominant factor. Figure 6k-n shows the
averaged contributions from individual and total emissions of BVOCs and soil NOx for
a north and south region marked respectively by purple and black boxes in Figure 6a&b.
The averaged ozone changes in the North and South region are in the range of -4~4
ppbv, and BVOCs and soil NOx on average contribute 28% to the total changes. The
combined contribution of BVOCs and soil NOx is more consistent with that of BVOCs,
and the soil NOx-induced changes are small in all cases except Northern China under
the weak WPSH conditions. The exception in Figure 6m might be due to the ratio of
VOC to NOx in the North region under weak WPSH conditions, which shifts towards
the NOx-limited regime, making ozone concentration more sensitive to the change of
NOx. In sum, the result emphasizes the role of BVOCs emission in total chemistry
production.




## 4.  Conclusions and Discussion


In this study, we highlight the role of weather systems like WPSH on surface ozone
pollution in China interpreted with a comprehensive mechanism analysis. Statistical
analysis of surface observation reveals a dipole-like ozone change associated with the
WPSH intensity, with stronger WPSH increasing surface ozone concentration over
Northern China but reducing it over Southern China, and a reversed pattern during its
weak phase. This phenomenon is associated with the change of meteorological
conditions induced by the change of WPSH intensity. Specifically, when WPSH is
stronger than normal, dry, hot south winds from inland area serves to increase
temperature in Northern China but decrease relative humidity, cloud cover, and
precipitation, creating an environment that is favorable for surface ozone formation. In
Southern China, the south winds transport ozone-poor air as well as water vapor from
the ocean, which dilute ozone and also increase relative humidity, cloud cover, and
precipitation, and decreases solar radiation. Opposite changes are found during weaker
WPSH conditions.

This dipole pattern of surface ozone changes is well reproduced by the GEOS-Chem
model simulations. Diagnosing the model budgets also suggests that chemistry serves
as the key process determining the direction of the ozone change, including both
changes in BVOCs and soil $NO_x$ emissions and the changes in chemical reaction rates



with the WPSH intensity. Ozone changes caused by natural emission (including BVOCs
and soil $NO_x$) accounts for ~30% of the total ozone changes.

As WPSH is associated with continental scale circulation patterns, such as the East
Asian Summer Monsoon (EASM), several previous studies also discussed the impact
of EASM on ozone pollution in China (Yang et al., 2014; Han et al., 2020). However,
our study differs from the EASM related ones in that (1) the EASM has complex space
and time structures that encompass tropics, subtropics, and midlatitudes. Given its
complexity, it is difficult to use a simple index to represent the variability of EASM
(Wang et al., 2008; Ye et al., 2019), whereas the location and definition for WPSH are
much definitive (Lu et al., 2002; Wang et al., 2012); and (2) The influences of EASM
on ozone mainly represent interannual scale as EASM indices are defined by
month/year, while the WPSH is a system more suitable to explore the day to day
variability ozone, which is meaningful for short-term ozone air quality prediction.

A better understanding of the internal mechanism of WPSH's impact on ozone air
quality can also help assess the air quality variation more comprehensively under
climate change. The location and intensity of WPSH keep changing over time, e.g.,
Zhou et al. (2009) demonstrated that WPSH has extended westward since the late 1970s,
and Li et al. (2012) indicated that North Pacific Subtropical High will intensify in the
twenty-first century as climate warms. Nonetheless, there still exists a great uncertainty
about how WPSH will change under climate change, and further studies are needed to





discuss the responses of ozone to synoptic weather systems like WPSH in future
scenarios. In addition, the variability of WPSH is found to be related to global climate
variabilities such as ENSO (Paek et al., 2019) and PDO (Matsumura et al., 2016).
Therefore, how natural climate variabilities like ENSO and PDO interact with WPSH
to impact ozone air quality also needs more investigation.

**Data and model availability**
All the measurements, meteorological data are accessible online through the websites
given above. The GEOS-Chem model is a community model and is freely available
([www.geos-chem.org](www.geos-chem.org)).

**Author contributions**
J.L. and Z.J. designed the study. Z.J. ran the GEOS-Chem model and performed the
analysis. X.L. and L.Z. helped in the GEOS-Chem simulation. C.G. and H.L. helped in
the budget diagnosis. Z.J. and J.L. wrote the paper. All authors contributed to the
interpretation of results and the improvement of this paper.

**Competing interests**
The authors declare that they have no conflict of interest.

**Acknowledgement**
We thank the China National Environmental Monitoring Centre for supporting the



nationwide ozone monitoring network and the website (http://beijingair.sinaapp.com/)
for collecting and sharing hourly ozone concentration data. We appreciate GMAO for
providing the GEOS-FP meteorological data
(http://ftp.as.harvard.edu/gcgrid/data/GEOS_2x2.5/GEOS_FP). We also acknowledge
the efforts of GEOS-Chem Working Groups and Support Team for developing and
maintaining the GEOS-Chem model.

**Financial Support**
This work is funded by the National Key Research and Development Program of China
(No. 2017YFC0212803) and Open fund by Jiangsu Key Laboratory of Atmospheric
Environment Monitoring and Pollution Control (KHK1901).

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





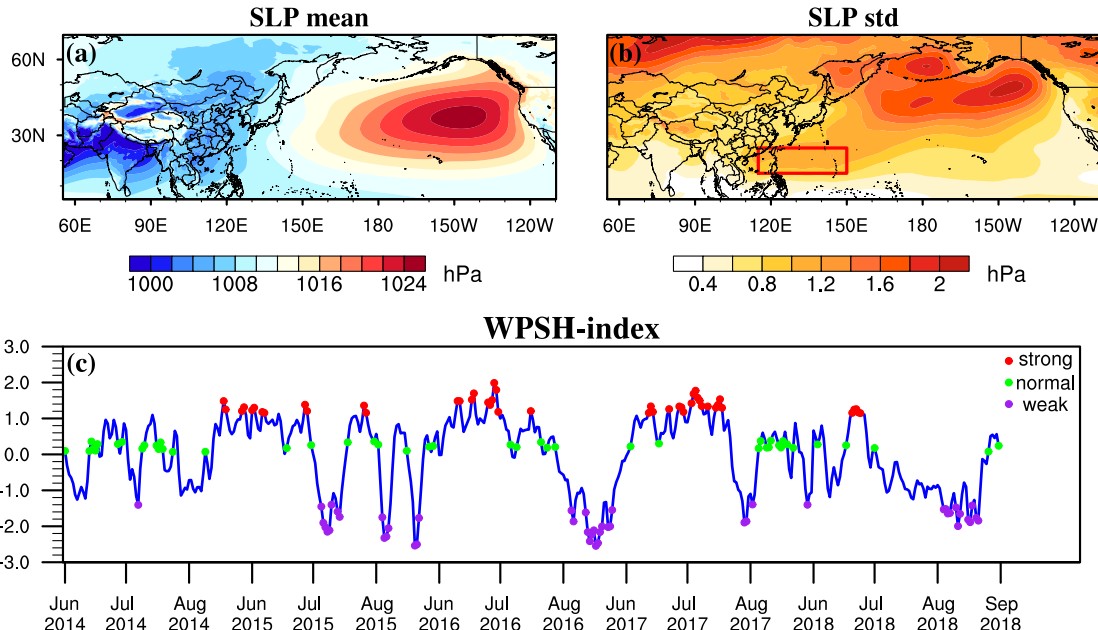

**Figure 1.** West Pacific Mean Sea Level Pressure (a) and its standard deviation (b), calculated using June, July, August (JJA) data from 1979 to 2018. Red box in (b) indicates the region (15-25°N, 115-150°E) used to calculate the WPSH-index. (c) shows the time series of WPSH-index and the selections of three types of WPSH. The blue line represents the normalized WPSH-index of 460 days in JJA from 2014 to 2018. Red dots represent strong WPSH days, green dots represent normal WPSH days and purple dots represent weak WPSH days.



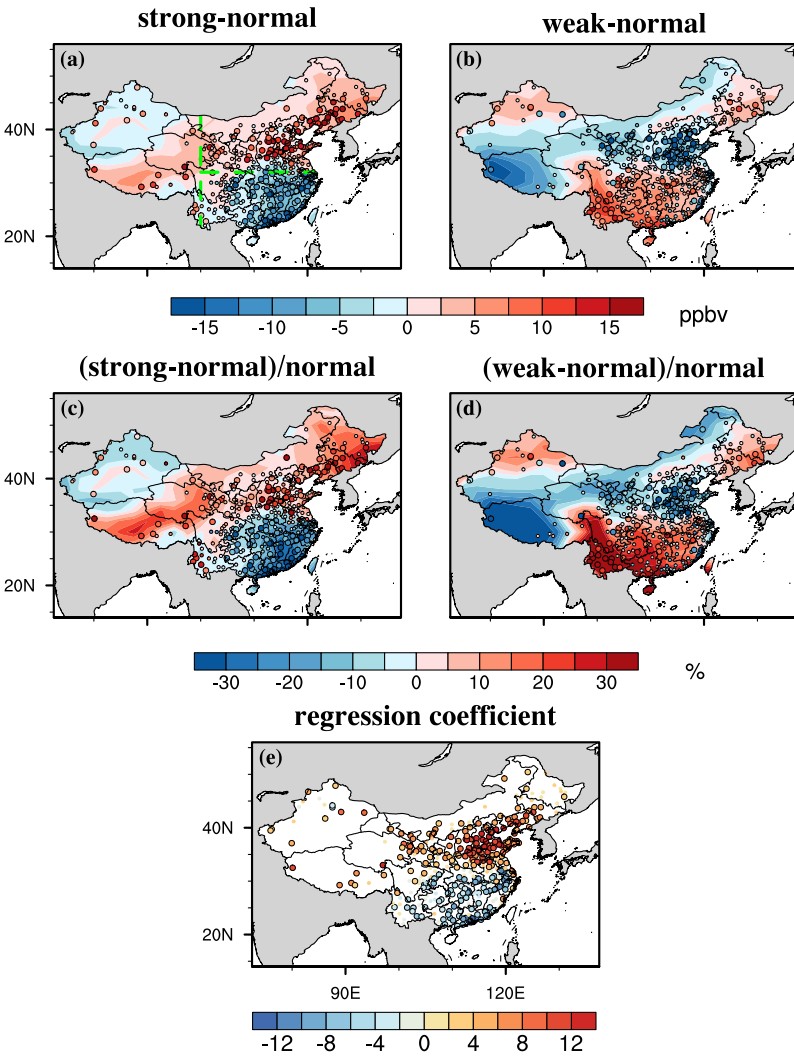

**Figure 2.** The observed (symbols) and simulated (filled contours) difference of MDA8 (ppbv) during strong and weak WPSH relative to normal WPSH days. (a) MDA8 of strong WPSH minus normal WPSH days, (b) MDA8 of weak WPSH minus normal WPSH days. (c) The percentage change of MDA8 of strong WPSH relative to normal, (d) the percentage change of MDA8 of weak WPSH relative to normal. (e) The regression coefficient between MDA8 in JJA from 2014 to 2018 and WPSH-index for cities in China. Larger dots with black circles in (a-e) are sites with significant level less than 0.05 from Student's *t*-test. The vertical green line in (a) is the boundary of Eastern China and the horizontal green line is the division of Northern and Southern China.

**Figure 3.** The difference of composite meteorological fields between different WPSH types. The first row corresponds to the difference between strong and normal WPSH days, and the second row correspond to the difference between weak and normal WPSH days. The meteorological variables including SLP, wind, precipitation, relative humidity, cloud cover, solar radiation, and 2 m temperature. The cross symbols indicate grids with significant levels less than 0.05 from Student's *t*-test.



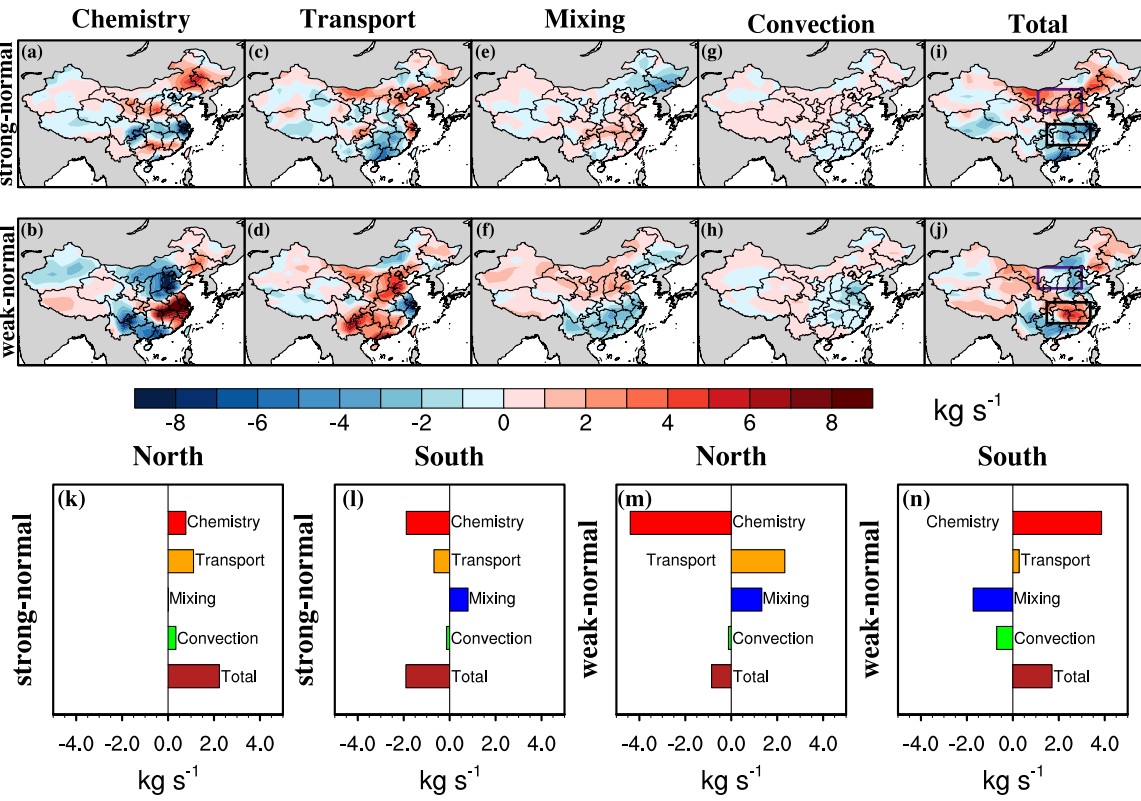

**Figure 4.** The budget diagnostics (kg s⁻¹) including chemistry, transport, mixing, and convection in the GEOS-Chem model. (a-j) The first row shows the differences between strong and normal WPSH days and the second row shows the differences between weak and normal WPSH days. (k-n) The area-averaged budget diagnostics (kg s⁻¹) for a north (36.0-42.0°N, 105.0-117.5°E) and south (26.0-32.0°N, 107.5-120.0°E) region (purple and black boxes in (i) and (j)).

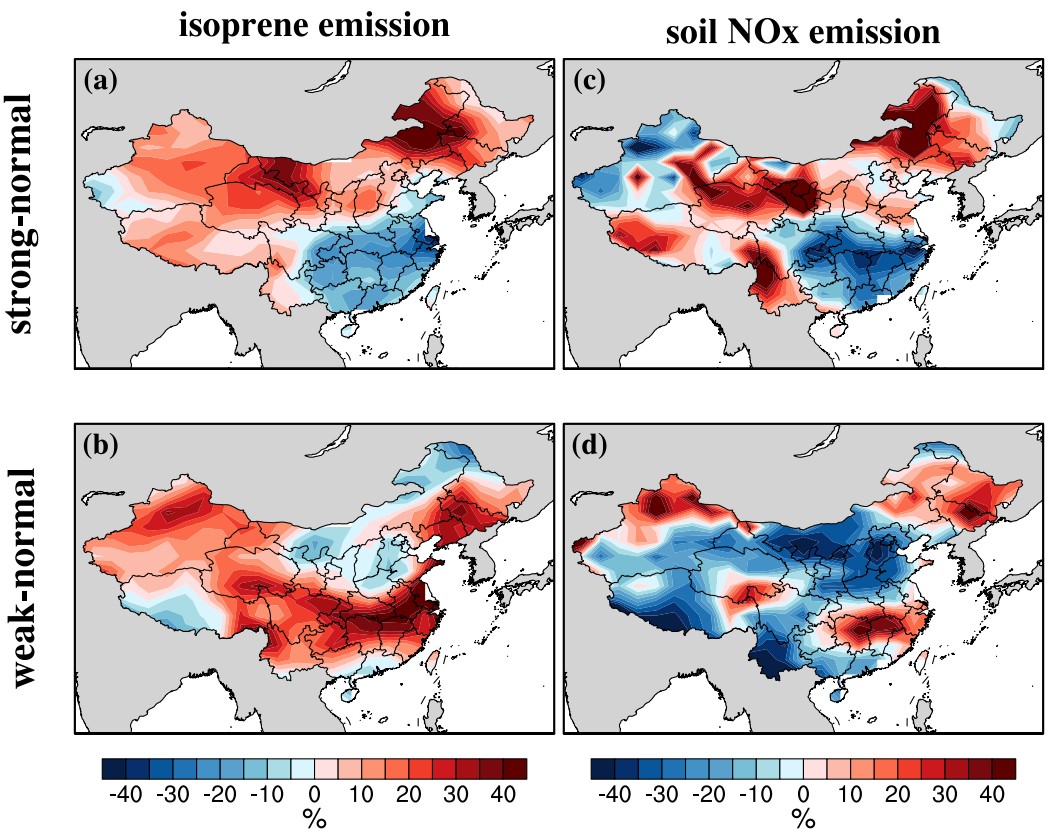

**Figure 5.** The changes of isoprene (a proxy of biogenic emission), soil NOx emission in GEOS-Chem model. The first row shows the relative differences (percentage) between strong and normal WPSH conditions and the second row shows those between weak and normal WPSH conditions.

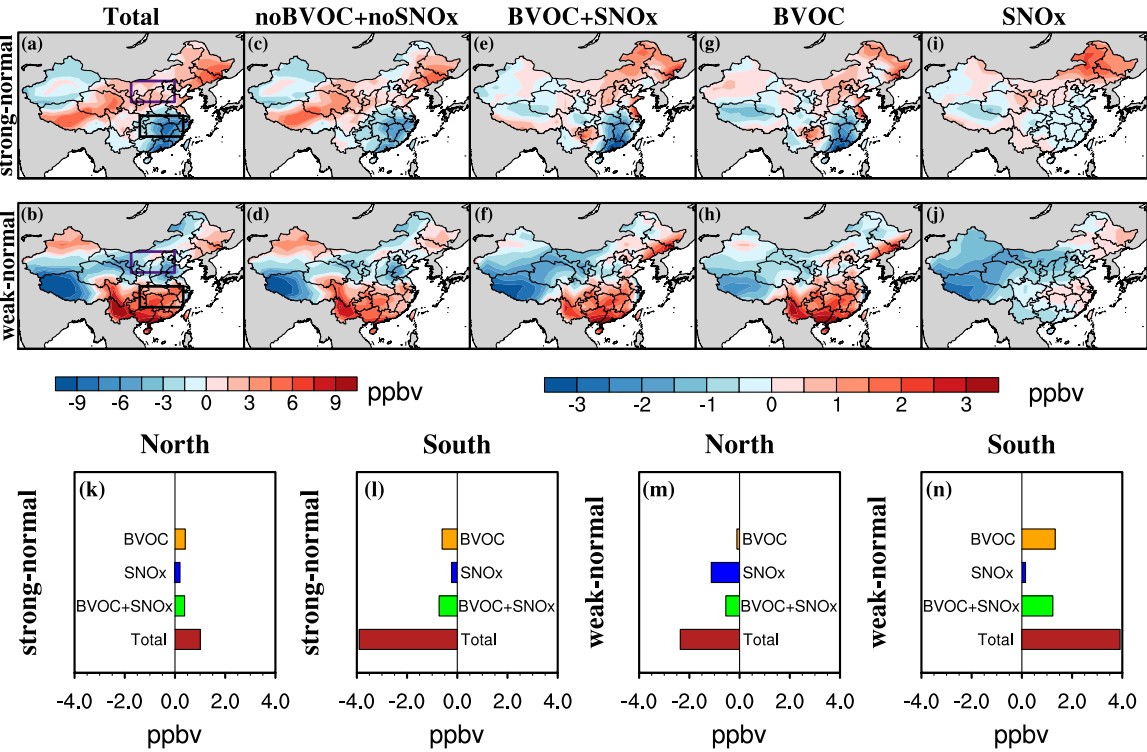

**Figure 6.** (a) and (b) show the simulated difference of MDA8 (ppbv) of strong and weak WPSH relative to normal WPSH, same as Figure 2a&b (filled contours). (c) and (d) are the same as (a) and (b) except turning off MEGAN and soil NOx emission. (e) and (f) show the difference between simulations with MEGAN and soil NOx emission on (Figure 6a&b) and off (Figure 6c&d), which represent the contribution of BVOC and soil NOx. (g) and (h) show the difference between simulations with MEGAN emission turned on and off, which represent the contribution of BVOC emission. (i) and (j) show the difference between simulations with soil NOx emission turned on and off, which represent the contribution of soil NOx emission. Note that (a-d) use the left colorbar and (e-j) use the right colorbar. (k-n) The contribution of BVOC, soil NOx (SNOx), BVOC together with soil NOx (BVOC +SNOx) for a north (36.0-42.0°N, 105.0-117.5°E) and south (26.0-32.0°N, 107.5-120.0°E) region (purple and black boxes in (a) and (b)).