# Peer review of "Impact of Western Pacific Subtropical High on Ozone Pollution over Eastern China"

_Atmospheric Chemistry and Physics, 2020_

## Referee Comment (RC1) · Anonymous Referee #2 · 4 Dec 2020

Review of "Impact of western pacific subtropical high on ozone pollution over eastern china"

This study presents a combined modelling and observational investigation of how meteorological conditions associated with the western pacific subtropical high (WPSH) affect surface ozone. The manuscript tells a nice story, with each piece of analysis following on from the previous. Their approach does represent a broader (temporally and spatially) and more coherent analysis than previous studies, particularly Zhao and Wang (2017).

The manuscript is well written and leads the reader through the analysis in a very clear manner, particularly the introduction. Observational analysis is backed up convincingly by a modelling study which seeks to determine the effect of natural emissions on ozone

variability. This modelling study further demonstrates the importance of physical and chemical mechanisms during different phases of the WPSH.

Main comments:

1) My main comment is about what this manuscript presents that isn't already published. To me it seems as if the manuscripts novelty is in the modelling, and improved understanding about the processes that alter the ozone budget under the WPSH regimes. However, in the conclusion and abstract much of the text is dedicated to drawing conclusions about ozone changes driven by meteorology, which is very similar to the work of Zhao and Wang (2017). I do note that the authors do point out that their study considers and observational record two years longer than Zhao and Wang. The paper provides useful insights from the modelling approaches, though my opinion is that these insights should be the focus of the paper.

2) The use of north and south China does not seem consistent throughout the manuscript. At L205 north/south is demarcated at 32N. Later, at L333, north and south regions are defined 36-42N and 26-32N respectively. Some clarity would be beneficial. The choice of the north and south region (L333) seems somewhat arbitrary and need more rationale, as many conclusions in section 3.4 rest on this choice, particularly those surrounding the contributions of BVOCs, soilNOx etc in figures 6(i-n).

Minor comments:

-L137 Is this definition of weak, normal, strong conditions common? If not, more rationale about these percentile choices is warranted.

-L229-233 This paragraph and the accompanying graphs really clearly and nicely demonstrate the meteorological effects. However, I don't agree that figure 3c shows a decrease in precipitation over northern china, at least not significantly. Figure 3c shows very little change to me.

-L283 Are the modelled strong/normal/weak values calculated from the same days as

the observations? A direct comparison as seen in Figure 2 would require this, but it is not clear to me that this is the case.

-Figure 1c requires an axis label dependent on your normalisation procedure. It is not apparent what form of normalisation has been performed

-Other figures. The quality of the figures is excellent, if a little small.

Technical corrections:

-L72 'some led' -> 'some that led'

-L103 should 'since' -> 'in'?

-L110 Should 'following' -> 'preceding'?

-L114 was -> were

-L235 'temperatures, less' -> 'temperatures and less'

-L429 'much' -> 'more'?

---

## Referee Comment (RC2) · Anonymous Referee #1 · 27 Dec 2020

Review of "Impact of western pacific subtropical high on ozone pollution over eastern china"

General:

This paper examines how much impact the variability of Western Pacific Subtropical High (WPSH) have on the surface ozone over East China. A combined modeling and observational approach reveals the impact quantitatively as well as the involved chemical and physical processes. The manuscript is clear and well written, and I believe that the quantitative analysis is very important for a better understanding of summertime air quality in China. However, there are some major points which have to be clarified and discussed further, as described below.

[Figure]

Major points:

In this study, the budget analysis of PBL ozone are performed using the diagnostics calculated in the GEOS-Chem model to investigate how and how much the variability of WPSH induces the changes in the summertime ozone over East China. However, the relationships with the meteorological conditions are not fully investigated, as pointed out below. Further analysis and discussions are needed.

1) The ozone dry deposition process should be also taken into consideration for the budget analysis, because the variability of the WPSH influences not only the four processes diagnosed here (i.e. chemistry, transport, mixing and convection) but more or less the dry deposition process.

2) There is lack of quantitative analysis to clarify which meteorological variables (solar radiation, temperature, RH...) are key factors that lead to the changes in ozone chemistry (i.e. chemical production/loss of ozone). Further analysis is needed to clarify this point.

3) Intensities of convective activities associated with WPSH variation are supposed to be very large. However, the large differences in convective activities between weak and strong WPSH only induces the small differences in PBL ozone, as you pointed out (Figure 4). It is required to explain the mechanism.

4) In Figure 1c, the absolute differences in the WPSH-index between weak and normal WPSH days (purple and green dots) are several times larger than those between strong and normal WPSH days (red and green dots). It is required to discuss how this asymmetry affects the later composite analysis.

Minor comments:

- L103 to L106: If there are a reference paper or technical report on the observation data used here, it should be cited.

- L114: should "for 2014-2018" -> "for 1979-2018"? (see Figure 1a and 1b).

[Figure]

- L114 to L115: If there are a reference paper or technical report on the "GEOS-FP database", it should be cited.

- L168: should "Cloud convetion" -> "Vertical transport due to convective transport"?

- L178: I suppose that "MEGAN and soil NOx emissions turned off" means "BVOC and soil NOx emissions are set to zero". Is it right?

- L209: What is the ratio of "cities with significant differences"? This information should be described.

- L251 to L252: "high-pressure center in Northwest Pacific is ... shifted slightly southward (Figure 3b)". The readers cannot know which the southward shift is slight or not, because the difference in SLP between strong (weak) and normal WPSH days is only showed in Figure 3a (3b). The SLP composite under strong (weak) WPSH days should be also depicted in Figure 3a (3b).

- L257: Does "abnormal changes" mean "asymmetric changes" in L254? Is it an appropriate expression in the context?
* * *

---

## Referee Comment (RC3) · Anonymous Referee #3 · 8 Jan 2021

This paper is studying the impact of Western Pacific Subtropical High (WPSH), a major synoptic system bringing specific meteorological conditions, on ozone over Eastern China in the summer months. It shows that when this system is strong, Northern China is seeing higher ozone compare to normal WPSH conditions. When the system is weak, Southern China is seeing higher ozone compare to normal WPSH conditions. Using the CTM GEOS-Chem, the authors show that chemistry (net chemical production = reaction rate and amount of ozone precursors) has a decisive role for ozone changes with respect to WPSH conditions. Natural emissions of precursors from biogenic and soil sources which are impacted by the temperature modulated by WPSH shows a non-negligible role to ozone changes.

The paper is investigating in more details the role of WPSH on ozone variability, com-

plementing the work of Zhao and Wang 2017, using the CTM GEOS-Chem, which is very much appreciated. The manuscript is well written and the figures well displayed. I am in favor of its publication after taking into account the following minor remarks.

Abstract:

L 19-20: The sentence implies that meteorological conditions is the main factor that controls ozone production when it is only one of several factors (emissions of ozone precursors, amount of ozone precursors, amount of other species such as PM2.5, chemical regimes, etc...). The authors mention it at the end of the abstract but it should be clear right in the beginning of the paragraph.

Introduction:

L. 48: The authors should add the following publications Mills et al. (2018) and Fleming and Doherty et al. (2018) from the Tropospheric Ozone Assessment Report (TOAR):

Tropospheric Ozone Assessment Report: Present-day tropospheric ozone distribution and trends relevant to vegetation. Mills G, Pleijel H, Malley CS, Sinha B, Cooper OR, Schultz MG, Neufeld HS, Simpson D, Sharps K, Feng Z, Gerosa G, Harmens H, Kobayashi K, Saxena P, Paoletti E, Sinha V, Xu X,. Elem Sci Anth. 2018;6(1):47. DOI: 10.1525/elementa.302.

Tropospheric Ozone Assessment Report: Present-day ozone distribution and trends relevant to human health. Fleming, Z.L., Doherty, R.M., von Schneidemesser, E., Malley, C.S., Cooper, O.R., Pinto, J.P., Colette, A., Xu, X., Simpson, D., Schultz, M.G., Lefohn, A.S., Hamad, S., Moolla, R., Solberg, S. and Feng, Z., 2018. Elem Sci Anth, 6(1), p.12. DOI: 10.1525/elementa.73.

Data and methods:

L. 141: The authors should further explain and detail the composite analysis.

Results:

L.212: Typo, change "," to "."

L.218: As already mentioned, the composite analysis should be further explained and detailed in the method section.

L. 263: The authors should clarify they interpretation of ozone enrichment and dilution from the wind anomalies (strong versus normal or weak versus normal WPSH).

L. 281: Does 0.57 translate a reliable model performance? It seems rather modest. The authors should give a range of reliable models and their performance in terms of correlation coefficients. That would guide readers who are not experts in models performances. Could the authors add the (normalized) mean bias as well? This more exhaustive evaluation for summer months would nicely complement the work on spring months in Ni et al. (ACP 2018) cited by the authors.

Ni, R., Lin, J., Yan, Y., and Lin, W.: Foreign and domestic contributions to springtime ozone over China, Atmos. Chem. Phys., 18, 11447–11469, https://doi.org/10.5194/acp-18-11447-2018, 2018.

L. 310: How do the authors conclude about dilution and accumulation of ozone based on maps of wind anomalies only? This statement deserves more details and/or references.

L.315: Did the authors mean "free troposphere"?

L. 356: Add "(see Section 3.3)" as it seems to refer to the findings above.

―――――――――――――――――――――――

---

## Author Comment (AC1) · 12 Jan 2021

**Response to the review of "Impact of western pacific subtropical high on ozone pollution over eastern china":**

**We thank the referee for the detailed and constructive comments. We respond to each specific comment below. The referee's original comments are shown in blue. Our replies are shown in black. The corresponding changes in the manuscript are shown in *Italic black*.**
* * *
**Anonymous Referee #1:**

**Review of "Impact of western pacific subtropical high on ozone pollution over eastern china"**

**General: This paper examines how much impact the variability of Western Pacific Subtropical High (WPSH) have on the surface ozone over East China. A combined modeling and observational approach reveal the impact quantitatively as well as the involved chemical and physical processes. The manuscript is clear and well written, and I believe that the quantitative analysis is very important for a better understanding of summertime air quality in China. However, there are some major points which have to be clarified and discussed further, as described below.**

**Major points: In this study, the budget analysis of PBL ozone are performed using the diagnostics calculated in the GEOS-Chem model to investigate how and how much the variability of WPSH induces the changes in the summertime ozone over East China. However, the relationships with the meteorological conditions are not fully investigated, as pointed out below. Further analysis and discussions are needed.**

1) **The ozone dry deposition process should be also taken into consideration for the budget analysis, because the variability of the WPSH influences not only the four processes diagnosed here (i.e. chemistry, transport, mixing and convection) but more or less the dry deposition process.**

Thanks for pointing out this problem. We acknowledged that dry deposition is also an essential process in tropospheric ozone pollution. However, as we used the non-local PBL scheme in our simulation, the dry deposition is included in the "mixing" term of the budget analysis. We have explained this problem in the main text. We also added the diagnosis of dry deposition flux and dry deposition velocity in the supplementary.

*[Main text, Lines 211-218] :*

*Dry deposition is not separately discussed in the budget diagnosis, as this process is included in mixing when using the non-local PBL mixing scheme. However, as it is an important process for ozone removal, we show the dry deposition flux and velocity at the surface level in the supplementary (Figure S2). It is found that dry deposition velocity appears spatially correlated with precipitation, i.e., higher precipitation generally corresponds to higher dry deposition velocity, whereas dry deposition flux is proportional to the change in ozone concentrations (Figure 2).*

[Figure]

**Figure S2.** The changes in dry deposition flux and dry deposition velocity at the surface level in GEOS-Chem model. The first row shows the differences between strong and normal WPSH conditions, and the second row shows those between weak and normal WPSH conditions.

2) **There is lack of quantitative analysis to clarify which meteorological variables (solar radiation, temperature, RH. . .) are key factors that lead to the changes in ozone chemistry (i.e. chemical production/loss of ozone). Further analysis is needed to clarify this point.**

Thanks for pointing out this problem. In this paper, our primary focus is the impact of the coordinated change of the entire meteorological field induced by the variation of the WPSH weather system on ozone. As we mentioned in the introduction (Line 71-72), the meteorological variables are interrelated. For example, an increase in cloud cover is associated with reduced solar radiation, it is thus difficult to isolate and to quantify the contribution of each variable separately. However, we admitted that it is important to investigate which meteorological variables are the key factors. We therefore attempt to address this problem by correlation analysis of ozone and each meteorological variable to explain this problem to our best extent, as shown in Figure S3.

*[Main text, Lines 281-290] :*

*Among these meteorological variables, RH, solar radiation, temperature, and meridional wind are mostly closely related to surface ozone concentrations (Figure S3). In particular, for Northern China, the highest correlation (positive) is found between ozone and temperature. For Central Southern China along the Yangtze River basin, ozone is most highly correlated with RH. Whereas for Southern China, wind speed and meridional winds seem to play the dominant role. The latter variable also shows reversed relationship with ozone for Northern (positive) and Southern China (negative), highlighting the different characteristics in regional transport of ozone pollution. The results of our correlation analysis are also consistent with previous studies (Jeong and Park, 2013; Zhang et al., 2015; Gong and Liao, 2019).*

[Figure]

**Figure S3.** Correlation coefficients, between simulated daily MDA8 ozone concentrations and meteorological variables including SLP, precipitation, relative humidity, cloud cover, solar radiation, 2 m temperature, wind speed, 10 m U wind, and 10m V wind calculated for the summer periods from 2014 to 2018.

3) **Intensities of convective activities associated with WPSH variation are supposed to be very large. However, the large differences in convective activities between weak and strong WPSH only induces the small differences in PBL ozone, as you pointed out (Figure 4). It is required to explain the mechanism.**

Thanks for pointing out this lack of clarity. We added a few sentences to explain this mechanism.

*[Main text, Lines 416-427] :*

*Convection only induces minor modulation to the total changes, generally less than ±1 kg s⁻¹ and negligible for some cases (Figure 4l&m). There are two possible reasons for this insignificant change. On the one hand, as ozone is insoluble in water, the large changes in convective activities associated with the WPSH variation may only exert minor effect in the ozone concentration through wet scavenging. Instead, it*

*influences ozone concentration by the vertical transport of ozone as well as its precursors, but the average magnitude of convective transport is about one order smaller than that of chemistry. On the other hand, previous studies show that the effect of convective transport of ozone alone is to reduce the tropospheric column amounts while the convective transport of the ozone precursors tends to overcome this reduction (Wu et al., 2007; Lawrence et al., 2003). As a result, changes in ozone are neutralized and the net effect is weak.*

**4) In Figure 1c, the absolute differences in the WPSH-index between weak and normal WPSH days (purple and green dots) are several times larger than those between strong and normal WPSH days (red and green dots). It is required to discuss how this asymmetry affects the later composite analysis.**

Thanks for pointing this out. We did notice this asymmetry. However, the meteorological changes associated with strong and weak WPSH appear much more symmetric. Therefore, this feature should not affect the ozone response much. We added the following discussions about this asymmetry in WPSH index.

*[Main text, Lines 355-360]* :

*Although the WPSH index exhibits an asymmetric feature, with the difference between weak and normal days much larger than that between strong and normal days, the responses of meteorological variables appear more symmetric (Figure 3). This thus leads to the more symmetric change in ozone concentrations (Figure 2). Therefore, we consider this asymmetric behavior in WPSH strength has negligible effect in the response of ozone pollution.*

**Minor comments:**

**- L103 to L106: If there are a reference paper or technical report on the observation data used here, it should be cited.**

Thanks for pointing out this problem, we cited the Chinese standard document for ozone observation data.

*[Main text, Lines 109-112]* :

*The ozone data follows the standard released by the Chinese standard document HJ 654-2013 (MEP, 2013) and the pollutant concentration data is available at https://quotsoft.net/air/. We downloaded hourly surface ozone concentration data for all sites from 2014 to 2018.*

**- L114: should "for 2014-2018" -> "for 1979-2018"? (see Figure 1a and 1b).**

We are sorry for this confusion. Here the GEOS_FP meteorological fields are from 2014 to 2018. The timespan for ozone analysis is from 2014 to 2018. However, in section 2.2, in order to define the WPSH index, we used SLP data from 1979 to 2018 to calculate its climatological mean state and the standard deviation (Figure 1a&b). This SLP data comes from the ERA5 reanalysis rather than GEOS_FP. We added explanations to make it clear. We also added a citation and acknowledgment of the ERA5 dataset.

*[Main text, Lines 131-134]:*

*We first used the long-term ERA5 reanalysis SLP data (Hersbach et al., 2019; https://cds.climate.copernicus.eu/) to determine the climatology and variability of SLP over the northwestern Pacific. Figure 1a shows the multi-year averaged summertime SLP field from 1979 to 2018, and Figure 1b shows its standard deviation.*

**- L114 to L115: If there are a reference paper or technical report on the "GEOS-FP database", it should be cited.**

Thanks for your advice. We cited the "File Specification for GEOS-5 FP" here.

*[Main text, Lines 118-121]:*

*Meteorological fields for 2014-2018 were obtained from the Goddard Earth Observing System Forward Processing (GEOS-FP) database (GEOS-FP file specification document, Version 1.0 (11 Jun 2013)), which is the current operational met data product from the Global Modeling and Assimilation Office (GMAO).*

*[Main text, Lines 684-685]:*

*Lucchesi, R., 2013: File Specification for GEOS-5 FP. GMAO Office Note No. 4 (Version 1.0), 63 pp, available from http://gmao.gsfc.nasa.gov/pubs/office_notes.*

**-L158: should "Cloud convection" -> "Vertical transport due to convective transport"?**

Yes, thanks for pointing this out. We have revised it accordingly.

**-L178: I suppose that "MEGAN and soil NOx emissions turned off" means "BVOC and soil NOx emissions are set to zero". Is it right?**

Yes, HEMCO has a list of emission extensions and the GEOS-Chem users can decide whether these emissions should be used or not. There are logical switches for all datasets listed in HEMCO_Config.rc to facilitate turning different datasets on/off.

"MEGAN and soil NOx emissions turned off" here means that these two emissions datasets are not read in. We rephrased our expression to make it clear.

*[Main text, Lines 203-206]* :

*We also performed another set of simulations with MEGAN and soil NOx emissions turned off to explore the contribution of natural emissions; in this case, these two emission datasets are not read in during the simulation.*

**-L209: What is the ratio of "cities with significant differences"? This information should be described.**

Thanks for pointing it out. We added the ratio in the main text.

*[Main text, Lines 250-259]* :

*Quantitatively, 45% and 31% of the cities show significant differences (p-value<0.05) in Student's t-test for the strong and weak WPSH relative to normal days, respectively. During strong WPSH days, the average MDA8 increased by 10.7 ppbv (+19%, Figure 2a&c) in Northern China and decreased by 11.2 ppbv (-24%, Figure 2a&c) in Southern China. Under weak WPSH conditions, the average MDA8 decreased by 10.2 ppbv (-17%, Figure 2b&d) in Northern China and increased by 4.6 ppbv (+10%, Figure 2b&d) in Southern China. This dipole change of ozone is also confirmed by a regression analysis of surface ozone against the WPSH index (Figure 2e),in which 71% cities show significant signals (p-value<0.05) with positive coefficients over Northern China and negative values in Southern China.*

**-L251 to L252: "high-pressure center in Northwest Pacific is . . . shifted slightly southward (Figure 3b)". The readers cannot know which the southward shift is slight or not, because the difference in SLP between strong (weak) and normal WPSH days is only showed in Figure 3a (3b). The SLP composite under strong (weak) WPSH days should be also depicted in Figure 3a (3b).**

We are sorry for this confusion. What we want to express is the difference of the SLP between normal and weak WPSH. We rephrased this sentence to eliminate this ambiguity.

*[Main text, Lines 304-305]* :

*Under the weak WPSH condition, it shows a negative anomaly center in the Northwest Pacific Ocean and to the southeast of China coast (Figure 3b).*

**- L257: Does "abnormal changes" mean "asymmetric changes" in L254? Is it an appropriate expression in the context?**

The "abnormal changes" here correspond to "solar radiation decreased and total precipitation increased in Guangdong province". We acknowledge that "asymmetric changes" is a more appropriate expression and we now used this word instead.

*[Main text, Lines 309-311]:*

*However, these asymmetric changes in meteorology well match the observed decrease of ozone in Guangdong province.*

---

## Author Comment (AC2) · 12 Jan 2021

**Response to the review of "Impact of western pacific subtropical high on ozone pollution over eastern china":**

We thank the referee for the detailed and constructive comments. We respond to each specific comment below. The referee's original comments are shown in blue. Our replies are shown in black. The corresponding changes in the manuscript are shown in *Italic black*.
* * *
**Anonymous Referee #2:**

This study presents a combined modeling and observational investigation of how meteorological conditions associated with the western pacific subtropical high (WPSH) affect surface ozone. The manuscript tells a nice story, with each piece of analysis following on from the previous. Their approach does represent a broader (temporally and spatially) and more coherent analysis than previous studies, particularly Zhao and Wang (2017). The manuscript is well written and leads the reader through the analysis in a very clear manner, particularly the introduction. Observational analysis is backed up convincingly by a modeling study which seeks to determine the effect of natural emissions on ozone variability. This modeling study further demonstrates the importance of physical and chemical mechanisms during different phases of the WPSH.

**Main comments:**

1) My main comment is about what this manuscript presents that isn't already published. To me it seems as if the manuscripts novelty is in the modeling, and improved understanding about the processes that alter the ozone budget under the WPSH regimes. However, in the conclusion and abstract much of the text is dedicated to drawing conclusions about ozone changes driven by meteorology, which is very similar to the work of Zhao and Wang (2017). I do note that the authors do point out that their study considers and observational record two years longer than Zhao and Wang. The paper provides useful insights from the modeling approaches, though my opinion is that these insights should be the focus of the paper.

Thank you very much for pointing out this problem. We revised the abstract and conclusion to emphasize our focus in modeling and diagnosis of the phsycial processes.

*[Abstract, Lines 18-39]:*

[revised manuscript text omitted]

**2) The use of north and south China does not seem consistent throughout the manuscript. At L205 north/south is demarcated at 32N. Later, at L333, north and south regions are defined 36-42N and 26-32N respectively. Some clarity would be beneficial. The choice of the north and south region (L333) seems somewhat arbitrary and need more rationale, as many conclusions in section 3.4 rest on this choice, particularly those surrounding the contributions of BVOCs, soilNOx etc in figures 6(i-n).**

Thank you for pointing out this problem. The different definition is mainly due to the difference spatial representation of site observation and model grids. We added the following explanations to clarify this issue.

*[Main text, Lines 227-235]:*

*Regarding the region definition in this study, because in section 3.1 and section 3.2 the calculations are all site-based (city-average), we applied a single latitude division line of 32°N to separate Northern and Southern China and a longitude division line of 100°E as a boundary for a rough definition of Eastern China (green lines in Figure 2a). In section 3.3 and later, the paper mainly focused on the model result analysis, which is gird-based (region-average); thus, we used a north region and a south region with the same size and shape to ensure their comparability. The principle we chose the north and south region is based on the principle of avoiding the influence of coastline and covering as much land area as possible.*

**Minor comments:**

**-L137 Is this definition of weak, normal, strong conditions common? If not, more rationale about these percentile choices is warranted.**

Thanks for pointing out this problem. We added explanations about the choice of this division standard.

*[Main text, Lines 149-157]:*

*Specifically, days with WPSH-index exceeding the 90th percentile of its distribution are classified as strong WPSH days, the 45th -55th percentile as normal WPSH days, and those below the 10th percentile as weak WPSH days (Figure 1c). There are two main reasons for the setting of this division standard: 1) using the 10% percentile range ensures that we have the same number of days during the summer from 2014 to 2018 for each type and enough sample (46 days for each type) for the composite analysis and statistical test; 2) the chosen of the percentile threshold is to maximize the difference between strong, weak and normal WPSH conditions in the time span of our study.*

**-L229-233 This paragraph and the accompanying graphs really clearly and nicely demonstrate the meteorological effects. However, I don't agree that figure 3c shows a decrease in precipitation over northern china, at least not significantly. Figure 3c shows very little change to me.**

Thanks for pointing out this problem. We agree that the decrease in precipitation over northern China is not significant. We changed the expression as below. However, relative humidity shows a coherent reduction over Northern China, so our conclusion remains intact.

*[Main text, Lines 272-277] :*

*As a result, Northern China exhibits a decrease in relative humidity (Figure 3e) and an increase in temperature (Figure 3k). Although the precipitation does not show significant changes, the decrease in cloud cover (Figure 3g) increases the near-surface solar radiation (Figure 3i) and can further change the photochemical reaction rates, which partly explains the increase of ozone concentrations here (Jeong and Park, 2013; Gong and Liao, 2019).*

**-L283 Are the modelled strong/normal/weak values calculated from the same days as the observations? A direct comparison as seen in Figure 2 would require this, but it is not clear to me that this is the case.**

Thank you for pointing out this lack of clarity. The modeled strong/normal/weak values were calculated from the same days as the observations. We added an explanation to make it clear.

*[Main text, Lines 344-347] :*

*Figure 2 (filled contours) shows the simulated MDA8 changes during strong/weak WPSH days with respect to normal days (a&b) and their relative changes (c&d). The simulated strong/normal/weak values were calculated from the same days as the observations.*

**-Figure 1c requires an axis label dependent on your normalization procedure. It is not apparent what form of normalization has been performed.**

Thank you for pointing out this lack of clarity. We added explanations to make it clear.

*[Main text, Lines 141-145]:*

*Here we adopted the same method to calculate the geopotential height anomaly and divided the anomaly time series according to its standard deviation to obtain a normalized WPSH index. Then we used this index to represent the strength and variability of the WPSH (Figure 1c).*

**-Other figures. The quality of the figures is excellent, if a little small.**

Thank you for pointing out the size problem. We acknowledge that some figures are a bit small, which is also our concern. However, as we want to show different variables or processes together to facilitate comparison, it's not easy to enlarge it due to the large number of subplots. We provided a landscape version for figure 4 and figure 6 to make them appear larger.

**Technical corrections:**

**-L72 'some led' -> 'some that led'**

Revised as suggested. Thank you.

**-L103 should 'since' -> 'in'?**

Revised as suggested. Thank you.

**-L110 Should 'following' -> 'preceding'?**

Thank you for pointing out this lack of clarity. The "following calculation" here does not refer to the preceding quality control but to the calculations in sections 3.1 and 3.2. We now moved it to the end of section 2 and discussed the calculations we did in more detail.

*[Main text, Lines 227-235]:*

*Regarding the region definition in this study, because in section 3.1 and section 3.2 the calculations are all site-based (city-average), we applied a single latitude division line of 32°N to separate Northern and Southern China and a longitude division line of 100°E as a boundary for a rough definition of Eastern China (green lines in Figure*

*2a). In section 3.3 and later, the paper mainly focused on the model result analysis, which is gird-based (region-average); thus, we used a north region and a south region with the same size and shape to ensure their comparability. The principle we chose the north and south region is based on the principle of avoiding the influence of coastline and covering as much land area as possible.*

**-L114 was -> were**

Revised as suggested. Thank you.

**-L235 'temperatures, less' -> 'temperatures and less'**

Revised as suggested. Thank you.

**-L429 'much' -> 'more'?**

Revised as suggested. Thank you.

---

## Author Comment (AC3) · 12 Jan 2021

**Response to the review of "Impact of western pacific subtropical high on ozone pollution over eastern china":**

We thank the referee for the detailed and constructive comments. We respond to each specific comment below. The referee's original comments are shown in blue. Our replies are shown in black. The corresponding changes in the manuscript are shown in *Italic black*.
* * *
**Anonymous Referee #3**

This paper is studying the impact of Western Pacific Subtropical High (WPSH), a major synoptic system bringing specific meteorological conditions, on ozone over Eastern China in the summer months. It shows that when this system is strong, Northern China is seeing higher ozone compare to normal WPSH conditions. When the system is weak, Southern China is seeing higher ozone compare to normal WPSH conditions. Using the CTM GEOS-Chem, the authors show that chemistry (net chemical production = reaction rate and amount of ozone precursors) has a decisive role for ozone changes with respect to WPSH conditions. Natural emissions of precursors from biogenic and soil sources which are impacted by the temperature modulated by WPSH shows a non-negligible role to ozone changes. The paper is investigating in more details the role of WPSH on ozone variability, complementing the work of Zhao and Wang 2017, using the CTM GEOS-Chem, which is very much appreciated. The manuscript is well written and the figures well displayed. I am in favor of its publication after taking into account the following minor remarks.

**Abstract:**

L 19-20: The sentence implies that meteorological conditions is the main factor that controls ozone production when it is only one of several factors (emissions of ozone precursors, amount of ozone precursors, amount of other species such as PM2.5, chemical regimes, etc. . .). The authors mention it at the end of the

**abstract but it should be clear right in the beginning of the paragraph.**

Thanks for pointing this out. We acknowledge that ozone production is also influenced by some other factors. We now also mentioned other factors at the beginning of the paragraph. However, as the meteorological conditions is the major focus of this study, we added discussions of its relevance with other factors rather than putting these factors in a juxtaposition structure.

*[Abstract, Lines 19-22] :*

*The formation of surface ozone pollution highly depends on meteorological conditions which are largely controlled by regional circulation patterns, which can modulate ozone concentrations by influencing the emission of the precursors, the chemical production rates, and regional transport.*

**Introduction:**

**L. 48: The authors should add the following publications Mills et al. (2018) and Fleming and Doherty et al. (2018) from the Tropospheric Ozone Assessment Report (TOAR):**

**Tropospheric Ozone Assessment Report: Present-day tropospheric ozone distribution and trends relevant to vegetation. Mills G, Pleijel H, Malley CS, Sinha B, Cooper OR, Schultz MG, Neufeld HS, Simpson D, Sharps K, Feng Z, Gerosa G, Harmens H, Kobayashi K, Saxena P, Paoletti E, Sinha V, Xu X,. Elem Sci Anth. 2018;6(1):47. DOI: 10.1525/elementa.302.**

**Tropospheric Ozone Assessment Report: Present-day ozone distribution and trends relevant to human health. Fleming, Z.L., Doherty, R.M., von Schneidemesser, E., Malley, C.S., Cooper, O.R., Pinto, J.P., Colette, A., Xu, X., Simpson, D., Schultz, M.G., Lefohn, A.S., Hamad, S., Moolla, R., Solberg, S. and Feng, Z., 2018. Elem Sci Anth, 6(1), p.12. DOI: 10.1525/elementa.73.**

Thanks for your suggestion. We have added these two references accordingly.

**Data and methods:**

**L. 141: The authors should further explain and detail the composite analysis.**

Thanks for pointing this out. We explained the detail of the composite analysis.

*[Main text, Lines 159-168]:*

*Composite analysis of observed and simulated surface ozone, meteorological variable as well as related model processes are performed based on these three types. We first calculate the composite mean of each variable for the 46 days of each WPSH type. As we focus on the ozone and meteorology differences induced by WPSH variation, we further calculated and discussed the difference of the composite mean between strong and normal WPSH as well as between weak and normal WPSH. The statistical significance of the difference is tested using the Student's-t test. We consider that the two composite means are statistically different if the test result is significant above 95% level. All figures except Figure 1 are displayed in the form of the differences between composite means.*

**Results:**

**L.212: Typo, change "," to "."**

Revised as suggested. Thank you.

**L.218: As already mentioned, the composite analysis should be further explained and detailed in the method section.**

We have added details of the composite analysis in the method section, and we answered this above.

**L. 263: The authors should clarify they interpretation of ozone enrichment and dilution from the wind anomalies (strong versus normal or weak versus normal WPSH).**

Thanks for pointing this out. We clarified this by adding the specific ozone change directions corresponding to the wind anomalies.

*[Main text, Lines 315-321]:*

*(2) the transport indicated by wind anomalies serves to enrich or dilute ozone concentration depending on the wind direction. Take Southern China as an example, the anticyclonic wind anomalies under strong WPSH tend to dilute ozone and the cyclonic wind anomalies under weak WPSH tend to enrich ozone, which is also confirmed in the budget analysis in section 3.4 below. Alternatively, this wind anomaly pattern drives an opposite change in ozone pollution over Northern China.*

**L. 281: Does 0.57 translate a reliable model performance? It seems rather modest. The authors should give a range of reliable models and their performance in terms of correlation coefficients. That would guide readers who are not experts in models performances. Could the authors add the (normalized) mean bias as well? This more exhaustive evaluation for summer months would nicely complement the work on spring months in Ni et al. (ACP 2018) cited by the authors.**

**Ni, R., Lin, J., Yan, Y., and Lin, W.: Foreign and domestic contributions to springtime ozone over China, Atmos. Chem. Phys., 18, 11447–11469, https://doi.org/10.5194/acp-18-11447-2018, 2018.**

Thanks for pointing out this problem. We acknowledge that the coefficient of 0.57 is a modest value; however, it is acceptable in terms of model simulations. As the coefficients always varied with many factors such as years and the number of sites, it is difficult to provide a definable range for a reliable model. What we can do is to compare these evaluating parameters with previous model studies, which we find are close to our results. We added the normalized mean bias of the summer seasonal mean surface ozone MDA8 in the supplementary (Figure S5); we also added discussions about model performance in the main text.

*[Main text, Lines 334-342]:*

*The spatial correlation coefficients (R) between the observed and simulated seasonal mean MDA8 concentrations for summers from 2014 to 2018 are 0.57, 0.59, 0.70, 0.81, and 0.81, respectively. The mean bias (normalized mean bias) between the observed and simulated seasonal mean MDA8 concentrations are in the range of 7.1-9.4 ppbv (13%-22%) for summers from 2014 to 2018 (Figure S5). These evaluation results are comparable to those reported in previous studies (Lu et al., 2019b; Ni et al., 2018), despite the slight differences due to differences in season and sampling, proving the confidence of using GEOS-Chem to simulate ozone concentrations.*

[Figure]

**Figure S5.** Normalized mean bias (%) between simulated and observed seasonal mean surface ozone MDA8 concentration (ppbv) over China for summer from 2014 to 2018 (a-e).

**L. 310: How do the authors conclude about dilution and accumulation of ozone based on maps of wind anomalies only? This statement deserves more details and/or references.**

Thanks for pointing this out. First, the budget change in Figure 4c and the maps of wind anomalies are mutually verified. We are not concluding about dilution and accumulation of ozone solely based on the maps of wind anomalies. Second, the correlation analysis with winds (shown below in the bottom row of Figure S3) also supports this conclusion. We added details in the main text to emphasize this point.

*[Main text, Lines 377-383]*:

*For strong WPSH, the change of ozone budget due to transport exhibits an asymmetric pattern with decreases in most parts of Southern China and increases over Northern and Northeastern China (Figure 4c). As the correlation analysis shows that ozone responds to meridional wind positively in the north and negatively in the south (Figure S3i), the changes in transport budget are consistent with the WPSH-induced wind anomalies (Figure 3a), which tends to dilute surface ozone in the south and enhance it in the north.*

[Figure]

**Figure S3.** Correlation coefficients, between simulated daily MDA8 ozone concentrations and meteorological variables including SLP, precipitation, relative humidity, cloud cover, solar radiation, 2 m temperature, wind speed, 10 m U wind, and 10m V wind calculated for the summer periods from 2014 to 2018.

**L.315: Did the authors mean "free troposphere"?**

Revised as suggested. Thank you.

**L. 356: Add "(see Section 3.3)" as it seems to refer to the findings above.**

Revised as suggested. Thank you.

---

## Author Response (AR2)

**Response to editor:**

**The editor' original comments are shown in red. Our replies are shown in blue.**
* * *
Dear Dr. Jiang and co-authors,

thank you very much for your detailed revisions and replies. Before final publication of your manuscript in ACP, there is one sentence in the text, I kindly ask you to clarify. In lines 422/423 you write "... , but the average magnitude of convective transport is about one order smaller than that of chemistry." This sentence sounds a bit "odd" as if you compare apples with eggs (convective transport with chemistry). Which quantity do you refer to? What do you mean with "chemistry". Please clarify this sentence.
Yours,

Patrick Jöckel
* * *
Dear editor,

Thanks for pointing out this problem. We now revised this sentence to make it clear.

*[Main text, Lines 422-423]:*

*……, but the average change of ozone budget due to convection transport is about an order of magnitude smaller than that due to chemical processes.*

Yours,
Zhongjing Jiang